# Genomic and Transcriptomic Insights into the Genetic Basis of Foam Secretion in Rice Spittlebug *Callitettix versicolor*

**DOI:** 10.3390/ijms25042012

**Published:** 2024-02-07

**Authors:** Xiao Zhang, Hong Chen, Xu Chen, Aiping Liang

**Affiliations:** 1Tianjin Key Laboratory of Conservation and Utilization of Animal Diversity, College of Life Sciences, Tianjin Normal University, Tianjin 300387, China; 2Key Laboratory of Zoological Systematics and Evolution, Institute of Zoology, Chinese Academy of Sciences, Beijing 100101, China

**Keywords:** comparative genomics, transcriptome, foam, malpighian tubule, *Callitettix versicolor*

## Abstract

Many animal species produce protective foams, the majority of which exhibit evolutionary adaptability. Although the function and composition of foams have been widely studied, the genetic basis of foam secretion remains unknown. Unlike most species that produce foam under specific situations, spittlebugs continuously secrete foams throughout all nymphal stages. Here, we capitalize on the rice spittlebug (*Callitettix versicolor*) to explore the genetic basis of foam secretion through genomic and transcriptomic approaches. Our comparative genomic analysis for *C. versicolor* and eight other insect species reveals 606 species-specific gene families and 66 expanded gene families, associated with carbohydrate and lipid metabolism. These functions are in accordance with the composition of foams secreted by spittlebugs. Transcriptomic analyses of malpighian tubules across developmental stages detected 3192 differentially expressed genes. Enrichment analysis of these genes highlights functions also revealed by our comparative genomic analysis and aligns with previous histochemical and morphological observations of foam secretion. This consistency suggests the important roles of these candidate genes in foam production. Our study not only provides novel insights into the genetic basis of foam secretion in rice spittlebugs but also contributes valuable knowledge for future evolutionary studies of spittlebugs and the development of pest control strategies for *C. versicolor*.

## 1. Introduction

Many animal species produce frothy secretions to safeguard themselves, their nesting sites, or their eggs [1]. Despite the diversity in the biological roles of these foams across species, the majority of them exhibit evolutionary adaptability [2]. For example, to protect offspring, female praying mantises (*Mantis religiosa*) secrete foams and lay eggs in them [3]. Similarly, many locusts (Orthoptera) lay their egg pods deeply in the soil and then envelop the eggs in foamy secretions to prevent them from desiccation. These foams allow beneficial, symbiont bacteria to proliferate [4]. The poplar sawfly, *Stauronematus compressicornis*, uses its salivary gland to excrete foam that can inhibit the feeding of other insects and occupy the food niche [5]. Previous research has predominantly focused on illustrating the function and composition of foams, with little exploration into the genetic basis of foam secretion [6,7,8].

Cercopoidea (Hemiptera), widely recognized as spittlebugs, are distinguished by their nymphs inhabiting white foam that looks like frothy spit on their host plants [9]. In contrast to the majority of animals that release foam under specific situations, the spittlebugs exhibit a distinct behavior of continuously secreting substantial quantities of foam throughout their nymphal stages (Figure 1a,b) [10]. This continuous foam production is achieved by their initial secretion via the malpighian tubules [7,11], which serve as the functional analogue of the vertebrate kidney [12], and the agitation of secretions using the hind feet in the air [7,11]. These foams exhibit a notable abundance of mucopolysaccharides and proteins, which provide adhesive features for the foams [13,14,15]. Ultrastructure observations reveal a significant decline in secretory activity within the cells of adult malpighian tubules as compared to those of the nymphs [13,16]. Specifically, while there are extensive numerous secretory vacuoles and rough endoplasmic reticulum in the epithelial cells of nymphal malpighian tubules, they are replaced by dissolving secretory granules and autophagic vacuoles in their adult counterparts [13,16].

The rice spittlebug (Figure 1b), *Callitettix versicolor* (Hemiptera: Cercopidae), is an important polyphagous pest predominantly feeding on rice and maize in southeast and east Asia [17]. However, the genetic basis of foam secretion in *C. versicolor* remains unknown. Comparative genomics is an effective way to explore candidate genetic mechanisms of species divergence and adaptation [18,19]. We have recently reported the genome assembly of *C. versicolor*, which is the first reference genome for a cercopid species [20]. This resource allows for a comprehensive screening of the candidate genes and pathways related to foam secretion in this pest species.

In this study, we leveraged cutting-edge genomic resources to perform comparative genomic analysis among *C. versicolor* (Hemiptera: Auchenorrhyncha: Cercopoidea) and eight other insect species that do not secrete foams. These selected species are phylogenetically related to *C. versicolor*, which could provide a framework for genomic comparisons aimed at identifying gene families associated with foam secretion. Hemiptera includes three suborders: Heteroptera, Sternorrhyncha, and Auchenorrhyncha. We selected representatives from each suborder based on the availability and quality of their genome assemblies and annotated gene sets, therefore controlling the robustness of our comparative genomic analyses [18,21]. *Nilaparvata lugens* and *Laodelphax striatellus* from Auchenorrhyncha, *Bemisia tabaci* and *Acyrthosiphon pisum* from Sternorrhyncha, and *Apolygus lucorum* from Heteroptera were consequently included in our analysis. *Frankliniella occidentalis* and *Pediculus humanus corporis* were used as outgroups based on previous knowledge of paraneopteran phylogeny [22]. *Locusta migratoria* was also included in our comparative genomic analysis to root all phylogenetic trees. In addition, we sequenced the transcriptomes of malpighian tubules to examine differences in gene expression between nymphal and adult *C. versicolor*. The objective of this study was to explore the genetic basis of foam secretion in *C. versicolor.* The results of this study could contribute valuable knowledge to the evolutionary studies of spittlebugs and to the development of pest control strategies for this pest species.

## 2. Results and Discussion

### 2.1. Gene Family and Phylogenetics of Rice Spittlebug

The genetic foundation of species-specific adaptive traits can be revealed by comparisons among related species at the genomic level [18,19,21]. Here, we capitalized on the chromosome-level genome assembly of *C. versicolor*, a dataset recently generated by us [20], as well as eight other well-curated insect genome assemblies, to infer orthologous gene families, reconstruct the phylogeny of these species, and explore the genetic basis of foam secretion in spittlebugs (Figure 1a,c). Among nine species compared in this study, six (*C. versicolor* [20], *N. lugens* [23], *L. striatellus* [24], *A. lucorum* [25], *A. pisum* [26], *B. tabaci* [27]) belong to Hemiptera, while the remaining three (*F. occidentalis* [28], *P. humanus corporis* [29], and *L. migratoria* [30]) are taxonomically outside, but still closely related to, Hemiptera (Figure 1c, Appendix A). We extracted 153,665 genes from all nine species, and a total of 105,522 genes were clustered into 19,099 gene families (Figure 1c and Figure 2a, Appendix A). The mean family size is six genes. *A. lucorum*, *A. pisum* and *C. versicolor* are the three species with the most abundant unique orthologs (Figure 2a). Among these gene families, 2321 were shared by all nine species (Figure 2a). 

We then performed genome-scale phylogenetic analysis and established a maximum-likelihood tree for these insect samples using 1272 single-copy orthologous genes shared by them (Figure 2b). The phylogenetic tree covered six hemipteran species, one thysanopteran species, one psocodean species and one orthopteran species. Our results showed that the two brown planthopper species, *N. lugens* and *L. striatellus* (Auchenorrhyncha: Delphacidae) clustered together, forming a sister group relationship with the rice spittlebug *C. versicolor* (Auchenorrhyncha: Cercopidae). The divergence time between the common ancestor of the two brown planthopper species and the rice spittlebugs was estimated to be 225.71 million years ago (Ma). The three species of the suborder Auchenorrhyncha formed a monophyletic group with *A. lucorum* (Hemiptera: Miridae). The divergence time between the common ancestor of *A. lucorum* and the three auchenorrhynchan species was estimated to be 260.03 Ma. In addition, the two species belonging to the suborder Sternorrhyncha are clustered together. The Sternorrhyncha clade diverged from the common ancestor of auchenorrhynchan and heteropteran species 293.77 million years ago. Generally, the six hemipteran species included in our analysis represented three suborders within Hemiptera. Our results showed that one species in the suborder Heteroptera and three species in the suborder Auchenorrhyncha formed a sister group, while the two species in the suborder Sternorrhyncha formed a sister group with the four species in the suborders Heteroptera and Auchenorrhyncha. This maximum-likelihood tree established by genomic data is consistent with the phylogeny hypothesis of Hemiptera previously inferred from molecular and morphological data [31,32,33].

### 2.2. Spittlebug-Specific and Expanded Gene Families Implicated in Foam Secretion

The specificity of gene families from particular phylogenetic clades reflects essential genetic foundations of novel characteristics shared by these species [19,21]. Similarly, the expansion of gene families throughout evolutionary history has been shown to be one of the major mechanisms underlying the adaptive evolution of specific species [18,34]. 

Our comparative genomic analysis among nine insect species detected 606 gene families that only exist in rice spittlebugs (Figure 2b). The Gene Ontology (GO) enrichment analysis conducted on these spittlebug-specific gene families revealed a significant overrepresentation in functions related to foam synthesis and transportation (Figure 2c). The foam secreted by spittlebugs is a production of air and viscous mixtures formed at the proximal end of the malpighian tubule [7,11]. Previous histochemical analyses of these viscous mixtures had revealed that their compositions were rich in carbohydrates, particularly in the highly polarized carbohydrates, composed of repeating disaccharide units, such as Glycosaminoglycans (GAGs) [14,35]. Intriguingly, our comparative genomic analysis found that spittlebug-specific genes were overrepresented in carbohydrate metabolic activities. Specifically, these genes were enriched in GO categories, such as “carbohydrate metabolic process”, “mannosyltransferase activity”, and “hyalurononglucosaminidase activity”. The latter referred to metabolic activities involving hyaluronan, a type of GAG [36]. In addition, genes related to the “Golgi cisterna membrane” were also enriched in spittlebug-specific gene families, coinciding with the fact that most GAGs require modification steps that take place in and around the Golgi apparatus [16,37]. The GO enrichment analysis also revealed enrichment of spittlebug-specific gene families in categories related to the transportation of secreta, including in “regulation of transmembrane transport”, “regulation of body fluid levels” and “rhythmic behavior”, which likely reflects the unique characteristics of foam transportation and splitting in *C. versicolor*. By comparing the rice spittlebug genome with closely related ones, 66 gene families were identified to be expanded (Figure 2a–c). Similar gene functions were also recovered by the GO enrichment analysis based on these expanded gene families, repeatedly highlighting the distinctive features of foam splitting in this particular species (Figure 2c). For example, gene families with functions related to lysozyme activity were significantly expanded in the *C. versicolor* genome. Lysozymes are essential for hyaluronan metabolism [38], the degradation of which is one of the sources contributing to the viscosity in secretions that form foam [14,16].

As a polyphagous species imposing substantial survival pressure on diverse food crops in Asian countries, *C. versicolor* feeds on a wide range of plants, such as rice, maize, wheat, sugarcane, and soybean [17]. In accordance with this, the characteristic of polyphagia was also recovered by our comparative genomic analysis (Figure 2c).

### 2.3. Transcriptomic Profiling of Malpighian Tubules across Developmental Stages Provides Insights into Foam Secretion

Foam secretion is the most well-known characteristic of *C. versicolor*. During the first and second instar nymphal stages, the foam secreted by these insects is small and dense. Subsequently, as the insect grows bigger during the third and fourth instar stages, the size of the secreted foam increases [10]. At the fifth instar nymphal stage, the secretion of foam reaches its peak, covering the whole body of the rice spittlebug (Figure 1a) [10]. Foam secretion is a behavior distinctively observed in nymphs, which ceases upon their transition into the adult stage. The principal organ responsible for foam production is the malpighian tubule. Rice spittlebugs undergo five instars, with the fifth instar identified as the optimal developmental stage for exploring the genetic basis of foam secretion. This is because the fifth instar’s similarity to the adult stage in both micromorphology and macromorphology [7,11]. This similarity, therefore, minimizes potential differences arising from malpighian tubule development, accentuating the distinctive features of foam secretion during this stage [13,16]. As such, we compared transcriptomic profiles of malpighian tubules in nymphs and adult rice spittlebugs to further explore the genetic basis of foam secretion in spittlebugs. 

During sampling of malpighian tubules for RNA-seq, we noticed that while the morphology of midguts differed dramatically between nymphs and adults, no evident difference was observed in the morphology of the malpighian tubules, hindguts, or their relative positions between these two developmental stages (Figure 3a,b). We used 120 spittlebugs for transcriptomic profiling of malpighian tubules (60 of fifth-instar nymph and adult, respectively). To avoid bias caused by individual differences, 20 individuals were pooled as one sample. A total of 126,084,938 clean reads were generated from six RNA-seq libraries, with an average data size of 6.27 Gb for each sample (Appendix A). More than 92% of the reads were successfully mapped to the spittlebug genome, suggesting the high quality of this RNA-seq dataset (Appendix A). We detected a total of 15,679 expressed genes in malpighian tubules, 2141 of which were not reported in the original official gene set [20]. Although the official gene set for *C. versicolor* was produced based on multiple sources, including transcriptome data, the malpighian tubules were not taken into account. These newly predicted genes may represent the ones that were specifically expressed in malpighian tubules. We, therefore, assigned function annotations to these newly predicted genes (Appendix A), which were included in the following analyses.

We calculated Pearson’s correlation coefficients to assess the relationships between different samples (Figure 3c). The resulting heatmap showed that samples from the same developmental stage were clustered together, with correlation coefficients *r*^2^ > 0.99, demonstrating that our sampling strategy of pooling multiple individuals successfully eliminated the potential bias caused by individual characteristics. The correlation coefficients between samples collected from two different developmental stages, however, exhibited substantial variation (all *r*^2^ < 0.04), highlighting a distinct change in the transcriptomic profiles of malpighian tubules during the transition from nymph to adult stages.

### 2.4. Differentially Expressed Genes Related to Foam Secretion

We examined the difference in gene expression related to foam secretion by comparing nymphal samples of malpighian tubules with their adult counterparts. There were 3192 genes differentially expressed among these developmental stages, 55% of which were upregulated in nymphal malpighian tubules. These differentially expressed genes (DEGs) were widely distributed across all 10 chromosomes, suggesting extensive regulatory impact accompanying the transition to adulthood (Figure 4a).

To summarize the function of these DEGs and understand their roles in spittlebug development, GO and Kyoto Encyclopedia of Genes and Genomes (KEGG) enrichment analyses were then performed (Figure 4b, Appendix A). The results indicated that genes exhibiting differential expression between nymphal and adult malpighian tubules primarily participated in processes associated with juvenile development. For example, Gene Ontology categories related to juvenile hormone metabolism (GO:0006718, GO:0006716, GO:0035049) were significantly enriched. In addition, the “insect hormone biosynthesis” pathway was significantly overrepresented in the DEG set (Figure 4b). These findings were consistent with prior observations, demonstrating the importance of these functions during the maturation process in insects. Remarkably, some functions highlighted by our comparative genomic analysis were also recovered in this DEG set. Specifically, both DEGs and spittlebug-specific gene families were enriched in the GO categories of “carbohydrate metabolic process” and “Golgi cisterna membrane” (Figure 2c and Figure 4b). Similarly, the genes associated with “UDP-glycosyltransferase activity” were found to be significantly over-represented in both the DEG set and expanded gene families of the rice spittlebugs (Figure 2c and Figure 4b). In addition, the KEGG pathway analysis for DEGs yielded similar results. Glucometabolic pathways, such as “glycosaminoglycan degradation” and “pentose and glucuronate interconversions” exhibited enrichment significance (Figure 4b,c). Furthermore, the functions attributed to these DEGs align well with prior knowledge regarding foam production and transportation in spittlebugs [14,15]. Taken together, the functional consistency observed between the results obtained at the comparative genomics level and those obtained from differential expression analysis at the developmental transcriptomics level strongly recommended the important roles played by these candidate genes.

The foam produced by spittlebugs contains various carbohydrates and lipids, which are essential for maintaining the stabilization of these foams [8]. The transportation of lipids, on the other hand, relies on *ApoD* (*Apolipoprotein D*) [39]. Here, we found 14 spittlebug-specific *ApoD* genes that were differentially expressed between nymphs and adults (Figure 4c). Lysozymes play a central role in the modulation of foam viscosity in spittlebugs. Our comparative genomic analysis revealed a substantial expansion of gene families associated with lysozyme activity (Figure 2c). Here, in our differential expression analysis, four genes encoding lysozymes were also detected as key candidates (Figure 4c), indicating their importance in the process of foam formation. 

We examined the expression patterns of these key DEGs by performing a series of qPCR experiments (Figure 4d and Appendix A). Specifically, ten candidates displaying high expression levels in nymphal malpighian tubules but not in their adult counterparts were randomly chosen for validation (Appendix A). The expression profiles obtained from qPCR experiments for all 10 DEGs were generally consistent with the patterns identified in our transcriptome data, indicating the reliability of our RNA-Seq results (Figure 4d and Appendix A). 

An avenue for future research is to investigate the expression patterns of these candidate genes across various developmental stages and tissues and to validate the functions of these genes through the application of RNAi and CRISPR technologies [40]. These future directions would benefit from improving the laboratory rearing techniques of *C. versicolor* [41] and establishing more pure-breeding lines of this species.

## 3. Materials and Methods

### 3.1. Spittlebug Collection and Rearing

Laboratory stocks of *C. versicolor* were initially obtained from a wild population in Mulinping Village, Hu’nan, China (28°12′26″ N 109°39′28″ E), sampled in 2019. This wild population was originally found as the pest insect of rice (*Oryza sativa* L.). The first-generation offspring from these wild individuals were used to establish the chromosome-level genome assembly for *C. versicolor* [20]. The samples used in this study were collected from the same laboratory stock, specifically the second-generation offspring derived from the wild spittlebugs. The nymphs were fed on the roots of rice seedlings, and adults were fed on the leaves of the rice seedlings. All spittlebugs were reared in plastic boxes placed in an artificial climate chamber following previously published methods [10,41].

### 3.2. Genomic Data Collection and Functional Assignment

The genome assembly and official gene sets for *C. versicolor*, as well as eight other insect species (Figure 1c), including *A. pisum*, *A. lucorum*, *B. tabaci*, *L. striatellus*, *N. lugens*, *F. occidentalis*, *P. humanus corporis*, and *L. migratoria*. They were collected from previous publications [23,24,25,26,27,29,30,42]. The detailed sources of these data are listed in Appendix A. We assigned putative functions to these predicted genes by comparing them with those in public databases. Specifically, the proteins of these genes were blasted (BLASTP, E < 10^−5^) [43] against the NCBI nonredundant protein (NR, 20200921) [44], TrEMBL (202005), and SwissProt (202005) databases [45]. The best hits were assigned as functional annotations. GO terms were then assigned using GO annotations (20200615) downloaded from the GO Consortium [46]. In addition, we used BlastKOALA (20191220) to assign Kyoto Encyclopedia of Genes and Genomes (KEGG) Orthology (KO) terms [47]. 

All GO and KEGG enrichment tests were conducted using clusterProfiler v3.14.0 [48]. We used the UpSetR [49] and ggplot 2 [50] packages in R (v4.3) to visualize the results of intersection and enrichment analyses, respectively.

### 3.3. Comparative Genomic Analysis

We performed ortholog clustering analysis to identify orthologous genes among nine insect species using OrthoFinder v2.4 [51] with DIAMOND [52] (-e 0.001). These gene families identified by OrthoFinder were then annotated by PANTHER v15 [53]. MAFFT v7.313 was employed to conduct multiple alignments of single-copy genes (-local pair --maxiterate 1000) [54]. We further used Gblocks to remove divergent and ambiguously aligned blocks, which can enhance the quality of phylogenetic reconstruction [55]. Based on 1272 single-copy genes across whole genomes, we used IQ-TREE (v1.6.11) and ModelFinder implemented in it to select the best model (LG + F + I + G4) and resolve the general phylogeny of these insects by establishing Maximum Likelihood trees [56,57]. Bootstrap values for the final phylogeny were obtained based on 1000 replicates. The divergence time was estimated using the MCMCTree program in PAML v4.9i [58], calibrated with constraints obtained from the Timetree database (*L. migratoria* and *N. lugens* [350–378 million years ago (Ma)], *B. tabaci* and *A. pisum* [245–351 Ma], *F. occidentalis* and *A. lucorum* [269–440 Ma]) [59]. Taking advantage of these results, CAFE v4.2 was then employed to test gene family expansion or contraction [60]. Gene families with both family-wide *p* values and Viterbi *p* values of <0.05 were identified as significantly changed. 

### 3.4. Sampling and Transcriptome Sequencing of Malpighian Tubules

To compare the transcriptomic profiles of malpighian tubules between nymphal and adult *C. versicolor*, we collected malpighian tubules of fifth-instar nymphs and adults, respectively, sequenced them, and compared their gene expression profiles. Specifically, spittlebugs were starved and briefly anesthetized prior to dissection. For adults, the wings and legs were removed first. The bodies were placed in 1.5 mL centrifuge tubes and then washed with 75% anhydrous ethanol three times (two minutes each), followed by washing with 0.1% sodium dodecyl sulfate (SDS) three times (two minutes each). The washed bodies were dissected in Petri dishes containing 0.65% NaCl solution. For each individual, the filter chamber, hindgut, midgut, malpighian tubule, and salivary gland were separated carefully. Malpighian tubules from 20 individuals were pooled as one sample for RNA extractions and sequencing (*n* = 20 individuals per pool). For each developmental stage, three biological replicates were produced (*n* = 3 pools per developmental stage). Total RNA was extracted from each sample using the RNAprep Pure Tissue Kit (Tiangen, China). The quantity and integrity of RNA samples were assessed using Nanodrop (Thermo Fisher Scientific, Waltham, MA, USA), Qubit (Thermo Fisher Scientific, Nieuwegein, NL, USA), and Aglient 2100 bioanalyzer (Agilent Technologies, Santa Clara, CA, USA) systems. RNA-seq libraries were constructed following manufacturer instructions. We then used Qubit, Agilent 2100, and quantitative PCR (qPCR) to examine the quality of these libraries. The qualified libraries were then sequenced on Illumina NovaSeq 6000 platform (Illumina, San Diego, CA, USA). We cleaned the raw sequencing data by trimming adapter sequences, removing reads containing ploy-N, and filtering low-quality reads [61]. All transcriptomic analyses were conducted based on these clean reads.

### 3.5. Transcriptomic Analyses

We mapped all clean reads generated by RNA-seq to the spittlebug reference genome [20] using HISAT2 (v2.1.0) [62]. StringTie (v1.3.4) [63] was then used to assemble the transcripts guided by the official gene set of *C. versicolor* and quantify the gene expression values for each sample. For the new genes predicted by StringTie [64], which were not reported in the official gene set of *C. versicolor* [20], only the ones that contained more than one exon and coded proteins with at least 50 amino acids were maintained. The functional annotations of these newly predicted genes were assigned based on the public databases, including NR, SwissProt, COG, KOG and KEGG (Appendix A). To measure the expression level of each gene, we calculated the FPKM values (Fragments Per Kilobase of transcript per Million fragments mapped reads) for all genes following the HISAT-StringTie pipeline [64]. For differential expression analysis, six samples were divided into 2 groups (i.e., the nymph group and the adult group). DESeq2 (v1.6.3, test = “Wald”,fitType = “parametric”) [65] was applied to perform the differential expression analysis between these two groups. All *p* values were adjusted using the p.fdr function in R to control the false discovery rate of multiple testing. The differentially expressed gene (DEG) was defined as a gene with an FDR-adjusted *p* value < 0.01 and a fold-change value > 2. 

### 3.6. Quantitative Real-Time PCR

We verified the expression patterns of key candidate DEGs by performing a series of quantitative Real-Time PCR (qPCR) experiments. Three RNA samples extracted from the fifth-instar nymphal malpighian tubules and three RNA samples extracted from adult malpighian tubules were used in these experiments. To transform RNA into single-stranded cDNA, the FastKing 1st Strand cDNA Synthesis kits (KR116, Tiangen, China) were used following the protocol provided by the manufacturer. The obtained cDNA was then diluted 10 times for use as the reaction template. We used the Applied Biosystems StepOne Real-Time PCR System (Applied Biosystems, Foster City, CA, USA) and the 2X SG Fast qPCR Master Mix kit (High Rox, B639273, BBI, ABI) to perform the qPCR experiments. Each 20 μL reaction mixture contained 2.0 μL cDNA, 0.4 μL forward primer (10 μM), 0.4 μL reverse primer (10 μM), 10 μL SybrGreen qPCR Master Mix and 7.2 μL ddH_2_O. The templates were amplified using the following program: an initial denaturation at 95 °C for 3 min and 45 cycles at 95 °C for 5 s and 65 °C for 30 s. Following a well-established bioinformatic workflow [66,67], we evaluated all expressed genes and chose *ANT* as the reference gene for our RT-qPCR analysis. This involved normalizing the read counts derived from RNA-seq, establishing a cut-off for excluding weakly expressed genes, and ultimately selecting the gene with the lowest coefficient of variation to serve as the reference gene [66,67]. Primers for all qPCR experiments were listed in Appendix A. The relative expression level of each gene was calculated using the 2^−ΔCt^ method [68].

## 4. Conclusions

Foam secretion, a phenomenon widespread among various animal species, is well investigated for its adaptive roles but poorly understood at the genetic level. In this study, we took advantage of the chromosome-level genome assembly of the rice spittlebugs (*C. versicolor*) and the well-established laboratory lines of this species to explore the genetic basis for its foam secretion. A total of 19,099 gene families were identified among six hemipteran species and three outgroup species (Figure 1). The maximum-likelihood tree, constructed from 1272 single-copy orthologous genes, grouped spittlebugs together with two planthoppers (Figure 2b), which is in accordance with previous studies [31,32,33]. Our comparative genomic analysis identified 606 species-specific gene families and 66 expanded gene families in *C. versicolor* genome, which are enriched for gene functions related to carbohydrate metabolism, lipid metabolism, and their transportation (Figure 2c). These results were consistent with previous studies that had shown the abundance of carbohydrates and lipids in foams secreted by nymphal malpighian tubules in other spittlebug species [13,16]. The genetic basis of foam secretion in *C. versicolor* was also investigated at the transcriptomic level. Although no evident morphology difference in malpighian tubules was observed between the fifth-instar nymphs and the adults, their transcriptomic profiles were distinct from each other (Figure 3). A total of 3192 DEGs were detected all over the whole genome (Figure 4a). The functional enrichment analysis performed based on these DEGs recovered similar key functions that were highlighted by our comparative genomic analysis (Figure 2c and Figure 4b, Appendix A). The alignment of these functions with foam production, the differential expression of genes associated with these functions, as well as the birth and expansion of gene families related to these functions, have collectively emphasized the important roles played by these candidate genes. Manual inspections and qPCR experiments further supported the observed expression patterns of genes associated with carbohydrate and lipid metabolism and transportation (Figure 4c,d), thereby illuminating their roles in the process of foam secretion. 

Taken together, the sequence data and analytical results presented in this study provide novel insights into the genetic basis of foam secretion in rice spittlebugs, thus improving our understanding of spittlebugs’ foam composition, production mechanisms, and secretion processes. This study not only provides potential objectives for future functional experiments in *C. versicolor* but also serves as valuable resources for studies on spittlebug evolution and the development of its control strategies.

## Figures and Tables

**Figure 1 ijms-25-02012-f001:**
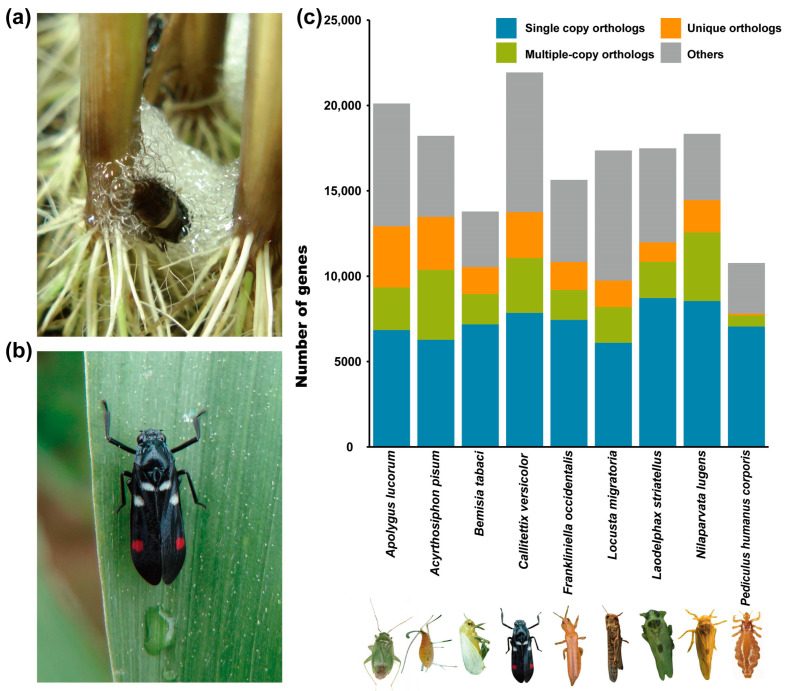
*Callitettix versicolor* and eight other insect species used for comparison. (**a**) The nymph of a *C. versicolor* covered itself with foam. (**b**) An adult *C. versicolor.* (**c**) Distribution of orthologous genes in *C. versicolor* and eight other insects used for comparative genomic analysis (photo credits listed in Appendix A).

**Figure 2 ijms-25-02012-f002:**
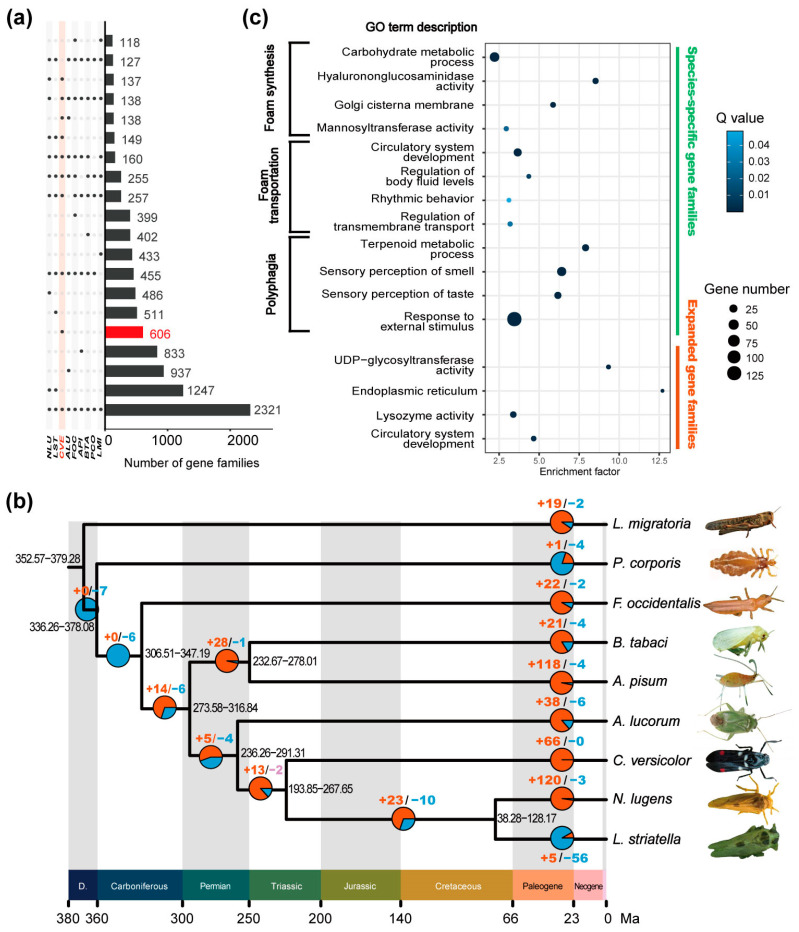
Comparative genomics of *C. versicolor* and other 8 insect species. (**a**) Intersection of gene families of nine insect species compared in this study. Sets of gene families that are part of the intersection are indicated by dark dots in the left matrix (CVE = *C. versicolor*. Abbreviations were listed in Appendix A). The number of shared gene families are shown with the bar graph. Top 20 intersections are shown. (**b**) Phylogenetic tree for *C. versicolor* and the other 8 insect species. Bootstrap values for all nodes are 100%. The *x*-axis represents divergence time (in million years ago, Ma) with units of geologic time shaded in distinct colors (D. = Devonian). The 95% confidence intervals of divergence time are indicated by black numbers near each node. The red and blue numbers represent the expanded and contracted gene families inferred by the CAFE program, respectively, which are also illustrated by corresponding pie charts below them. (**c**) The Gene Ontology (GO) enrichment analysis of the species-specific (blue) and expanded (red) gene families of *C. versicolor*. GO categories related to known characteristics of spittlebugs were shown. (all FDR-adjusted *p* (i.e., *Q*) < 0.05, Statistical details are provided in Appendix A).

**Figure 3 ijms-25-02012-f003:**
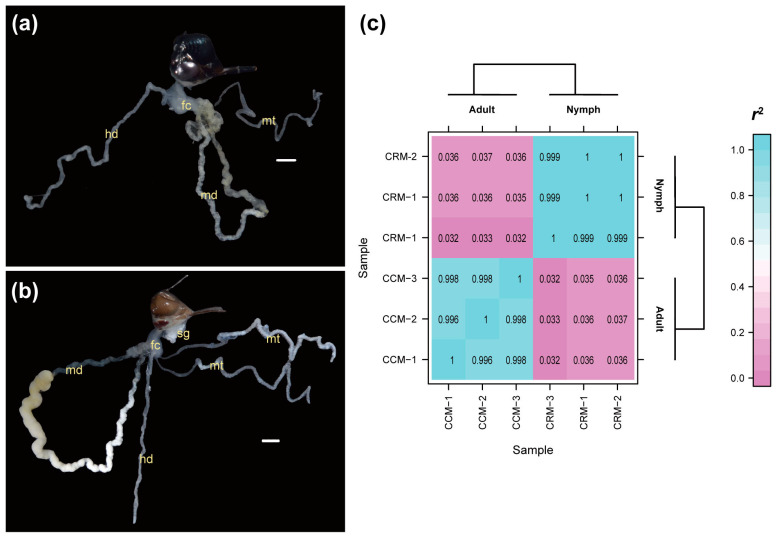
Morphological and transcriptomic profiles of malpighian tubules in different developmental stages of *C. versicolor*. (**a**,**b**) Morphology of malpighian tubules in the (**a**) fifth-instar nymph and (**b**) adult stages of *C. versicolor*. For comparison, the relative position shown in the figure was rearranged after dissection, and only two malpighian tubules were retained. Abbreviations: fc, filter chamber. hd, hindgut. md, midgut. mt, malpighian tubule. sg, sliver gland. Scale: 1 mm. (**c**) Correlation heatmap showing the relationships of transcriptomic profiles among six samples. The color scale refers to the value of Pearson’s correlation coefficients (*r*^2^). The sample IDs were labeled in the figure. Comprehensive details for these IDs can be found in Appendix A.

**Figure 4 ijms-25-02012-f004:**
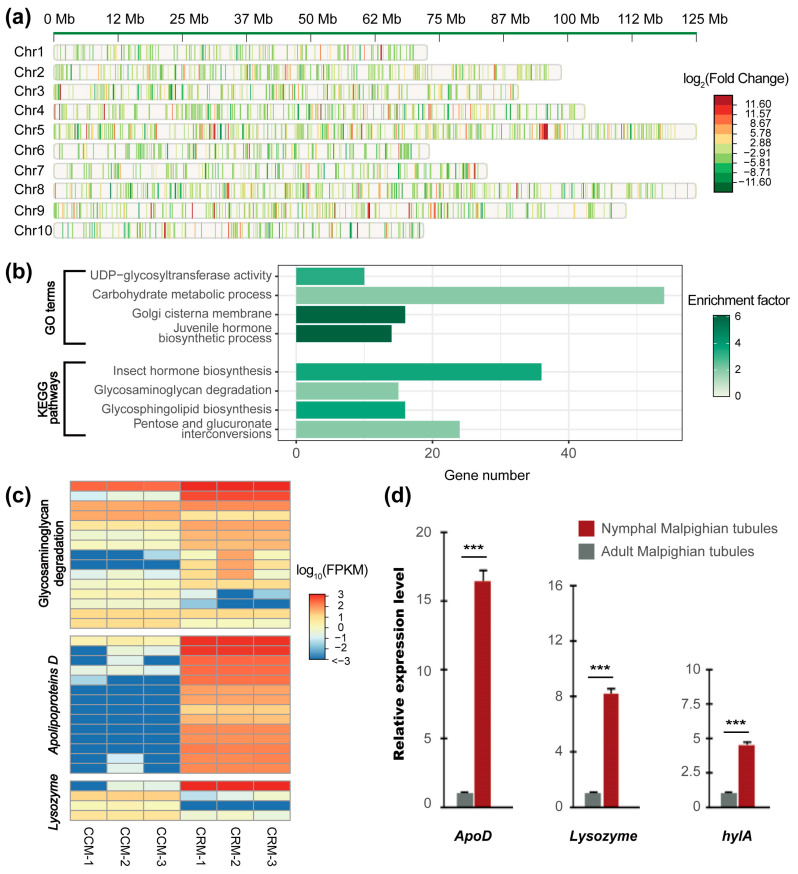
Genes significantly differentially expressed between nymphal and adult malpighian tubules. (**a**) Genomic landscape of differentially expressed genes (DEGs). The DEGs in the malpighian tubules of fifth-instar nymphs versus adults are plotted as bars. Colors indicate the log_2_ fold-change values, which show the magnitude of upregulation (red) versus downregulation (green). The chromosome numbers (1–10) are noted on the left side of the panel. (**b**) Functional profiles of DEGs. Labels show GO function and KEGG pathway categories significantly enriched in DEGs. All FDR-adjusted *p* ≤ 0.05. Statistical details are provided in Appendix A. (**c**) Heat map showing the express profiles of 33 DEGs belonging to the glycosaminoglycan degradation pathway, *Apolipoproteins D* gene family and *Lysozyme* gene family, respectively. The expression level of each gene was plotted for the nymphal (CRM) and adult (CCM) stages. (**d**) Validation of RT-qPCR for 3 key candidate DEGs. *** All *p* values < 0.001, *n* = 3 for each group.

## Data Availability

Raw RNA-seq reads for this study have been deposited in the National Center for Biotechnology Information (NCBI) Sequence Read Archive (SRA) under BioProject accessions, PRJNA789179. Genome assemblies used in this study was obtained from public database (Appendix A).

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
