# Peer review of "Genomic and Transcriptomic Insights into the Genetic Basis of Foam Secretion in Rice Spittlebug Callitettix versicolor"

_ijms, 2024, doi:10.3390/ijms25042012_

Round 1
Reviewer 1 Report
Comments and Suggestions for Authors
1. General comments:
Figure1c the reason to choose the eight other insects should explain in introduction, except they have gemomic data.
Line 369-398 reference gene was not scientific, they should be evaluated by GeNorm, NormFinder, BestKeeper and RefFinder etc.
Why not do transcriptome analysis of each instar nymph?
Are the unique and specific functional genes in C. versicolor found through the comparative genomics result, also in the transcriptome? Should be listed in the manuscript.
2. Specific comments
Introduction
Make sure “Malpighian tubules” not capital letter
Line 43 delete Figure 1a
Line 61 in Southeast and East Asia, only Asia is capital letter
Results and Discussion
Line 84 “the order Hemiptera” change as “Hemiptera”, the same as line 86
Line 87 figure1b right?and also 1c?
the order of Figure 2a and 2b is wrong, the same as follows. Adjust the order of figures.
Where figure 1b in manuscript? Also not in the right order.
Line 88 The mean family size is six genes. Is it right?
Hemipteran , Thysanopteran, Auchenorrhynchan etc should be hemipteran, thysanopteran, auchenorrhynchan. Also check all the manuscript.
Wrong font format “Line 96 the”, and”line 98 C. versicolor”
Delete Hemiptera: in line 96 and 98, because line 94 has said it.
Line 111 and 112,how about the result of morphology and molecular phylogenetic tree of these species?
Line 155 figure2a was wrong
Line 168-170 added the reference, besides the size of secreted foam increase may because the size of C. versicolor bigger from first to fifth instar nymphal stage?
Line 171 added the before fifth
Line 181 “Malpighian tubules (60 of each developmental stage; fifth-instar nymph and adult)” change as “malpighian tubules (60 of fifth-instar nymph and adult respectively)”
line 194 to 200 explain what’s the mean of CCM-1 to 3 and CRM-1 to 3.
What’s the different between DE and DEG and what’s DEG?
Figure4d just showed three genes, but line 260 say 10 genes?
Conclusion should be rewrite, don’t showed the method.
Materials and Methods
Line 314-316, genus name should be abbreviation
Line 386 “nymphal Malpighian tubules and three RNA samples” which instar are they?
Comments on the Quality of English Language
Moderate editing of English language required
Reviewer 2 Report
Comments and Suggestions for Authors
Dear Authors,
I read your submitted article titled "Genomic and transcriptomic insights into genetic basis of foam secretion in Callitettix versicolor (rice splittlebug)"and I consider it a very interesting contribution to the knowledge of genetic basis of the production of foam secretion produced by such insects , i.e.C.versicolor (rice spittlebug) through genetic and transcriptome approach. The impressive results are the identification of 606 species-specific gene families and 66 expanded gene families associated with carbohydrate and lipid metabolism, and the revealing, in the target species and other eight insect species studied , through the transcriptome analyses of Malpighiani tubules , of 3,192 differentially expressed genes. The mns is well presented and written in understandable English language, also using appropriate technical words. I only noticed a few minor errors ( missing geographical coordinates of collecting insect site and typing latin names of species recorded in the references list). See ,please, the attached revised word file.

Round 2
Reviewer 1 Report
Comments and Suggestions for Authors The manuscript has been sufficiently improved.Comments on the Quality of English Language
Minor editing of English language is required.
Author Response
We are grateful for the positive assessment and thoughtful suggestions. Regarding the English editing issue, we have found a colleague from the United States to help us proofread the manuscript. Following his feedback, we have edited our manuscript very carefully.